# An Ultra-Low-Power High-Precision Temperature Sensor Using Nonlinear Calibration with an Inaccuracy of +0.6/−1 °C from −30 °C to 90 °C for RFID Applications

**DOI:** 10.3390/s25092911

**Published:** 2025-05-04

**Authors:** Hanyang Wang, Zhonghan Shen, Hao Min

**Affiliations:** 1State Key Laboratory of Integrated Chips and Systems, Fudan University, Shanghai 200120, China; 21112020108@m.fudan.edu.cn; 2Chip Design Quanray Electronics Co., Ltd., Shanghai 200120, China; zhonghan.shen@quanray.com

**Keywords:** ultra-low power, nonlinear calibration, high-precision temperature sensor, RFID tag

## Abstract

This paper proposes a three-point nonlinear calibration scheme for an ultra-low-power, high-precision temperature sensor to address the issue where the temperature error of a 0.8 μW sensor exceeds ±1 °C in RFID (Radio-Frequency Identification) temperature measurement systems. The proposed calibration scheme introduces a temperature-dependent nonlinearity coefficient to the traditional linear calibration, effectively compensating for the sensor’s nonlinear output characteristics. To minimize calibration costs, a scheme embedding the calibration algorithm into the reader is proposed, along with a dichotomy-based approach for efficient temperature calibration. The experimental results demonstrate that, within the temperature range of −30 °C to 90 °C, the temperature error of five sensor samples can be reduced from ±8 °C to between −1 °C and 0.6 °C. This solution has been successfully implemented in mass production.

## 1. Introduction

Temperature sensing is a critical function in numerous applications, ranging from industrial process control to environmental monitoring. Over the years, various temperature sensing technologies have been developed, each with its unique characteristics, strengths, and limitations.

Thermocouples are widely used for their simplicity, ruggedness, and ability to measure a broad temperature range. However, they suffer from low sensitivity, require external power for signal generation, and exhibit nonlinear output characteristics, which can complicate calibration and signal processing.

Thermistors offer high sensitivity and fast response times, making them ideal for applications requiring quick temperature measurements. However, their nonlinear resistance–temperature relationship necessitates complex compensation techniques, and they are typically limited to lower temperature ranges compared to thermocouples.

SAW (Surface Acoustic Wave) devices leverage the piezoelectric effect to measure temperature changes. They are compact, lightweight, and offer good sensitivity. Nevertheless, their performance can be influenced by environmental factors such as humidity and mechanical stress, and they may require specialized signal processing.

Compared to thermocouples, thermistors, and SAW devices, RFID temperature sensing tags offer several significant advantages. First, they enable non-contact measurement, as data can be transmitted wirelessly without requiring direct contact with the object or environment being measured. This feature provides a distinct advantage in complex or hazardous environments, particularly in applications requiring long-distance or non-destructive detection. Additionally, RFID temperature sensors are more suitable for long-term use in harsh environments or industrial settings, as they do not necessitate complex installation or maintenance processes. Furthermore, RFID technology demonstrates cost-effectiveness, as the tags are reusable and generally have lower deployment costs compared to thermistors and SAW devices, especially in scenarios requiring high precision or complex structures. The flexibility of RFID temperature sensors allows them to be easily integrated into objects of various shapes and sizes, making them adaptable to diverse application scenarios. In contrast, thermocouples, thermistors, and SAW devices may face challenges in installation when dealing with specific structures or shapes of objects. In summary, RFID temperature sensors exhibit clear advantages in terms of wireless communication, ease of installation, cost-effectiveness, and remote monitoring capabilities. These benefits have contributed to their widespread adoption in fields such as industrial automation, logistics, healthcare, and smart home applications.

A prominent characteristic of RFID systems is their passive operation, which imposes stringent power consumption constraints on on-chip temperature sensors. Additionally, minimizing the chip area is essential for achieving cost-effective system design. Conventional on-chip temperature sensors generally operate by converting temperature variations into analog signals, which are subsequently digitized using an analog-to-digital converter (ADC). Among widely adopted methodologies, voltage-based and frequency-based conversion techniques are predominant [1,2]. Furthermore, in certain RFID-specific applications, temperature sensing is implemented through temperature-dependent changes in radio-frequency (RF) impedance, as referenced in [3,4].

Voltage-based conversion provides operational flexibility across a wide temperature range, as the signal can be amplified or attenuated using resistors or capacitors, thereby enabling precise temperature measurements. However, this method necessitates the integration of an ADC, which increases the circuit’s complexity. Additionally, high-resolution ADCs are known to consume more power and require more chip area.

Frequency-based conversion is a widely adopted method in RFID systems, offering low power consumption and relatively high accuracy by producing temperature-dependent output frequencies. This approach effectively balances precision and circuit complexity, making it suitable for resource-constrained applications. However, its operational sensing range is constrained by the limited linearity of the frequency response curve, which may restrict its applicability in certain scenarios.

Temperature sensing via RF impedance variation is characterized by ultra-low circuit complexity and minimal area consumption, rendering it particularly suitable for highly constrained applications. Nevertheless, this method is hindered by the inherent difficulty in achieving precise temperature quantification through impedance variation. Additionally, such techniques are prone to interference from external factors, including antenna detuning, environmental humidity, and fluctuations in the RF field strength, which may degrade measurement stability and accuracy.

The temperature sensor architecture proposed in this work employs a voltage-based approach that transduces temperature-induced voltage variations into charge, thereby incorporating the ADC operation into a unified and energy-efficient scheme. This paper presents a three-point nonlinear calibration scheme for ultra-low-power temperature sensors, achieving a measurement accuracy of −1 °C to 0.6 °C over a temperature range from −30 °C to 90 °C. Through exploiting the characteristics of RFID systems, the calibration algorithm is delegated to the reader side, thereby facilitating precise temperature measurements over an extensive range and reducing on-chip power and area overhead.

Section 2 examines ultra-low-power temperature sensors and presents the proposed calibration method. Section 3 details the circuit architecture. Section 4 presents comparative test results between traditional and nonlinear calibration methods. Finally, Section 5 explores potential enhancements to the calibration scheme and circuit design.

## 2. Proposed Nonlinear Calibration

The proposed temperature sensor architecture employs a three-point nonlinear calibration approach to reduce high-temperature nonlinear errors, thereby expanding the sensor’s measurement range and accuracy. Temperature-to-digital conversion involves three key steps: generating a temperature-dependent analog signal (ideally linear), transferring the analog signal to the ADC, and converting the signal via the ADC. In RFID temperature measurement applications, low power consumption is prioritized, followed by accuracy and speed, with overall energy efficiency being the most critical parameter. Consequently, a dual-integral ADC is adopted as the optimal solution.

Traditional dual-integral ADCs typically use voltage inputs, whereas the proposed temperature sensor (Figure 2 in [5]) utilizes current inputs, inducing voltage changes on an integration capacitor. This approach mitigates voltage sampling errors and noise interference, while significantly reducing overall power consumption, thereby enhancing temperature measurement energy efficiency.

Conventional calibration schemes, such as one-point and two-point calibration, rely on sensor output linearity. The proposed three-point nonlinear calibration is derived by analyzing potential error sources in various circuit stages, identifying three primary non-ideal factors: (1) leakage from the transistor, charge–discharge switch, and startup circuit; (2) input offset of the operational amplifier; (3) current mirror errors, particularly due to $V_{DS}$ variations in the charge–discharge section.

These issues result in a charging current with a nonlinear positive temperature coefficient, necessitating recalibration. By reformulating the temperature-to-digital conversion (T to $N_{out}$), we derive a multi-term equation incorporating linear and nonlinear calibration coefficients. The detailed derivation process is as follows.(1)$$V_{BE}=\frac{kT}{q}\ln\frac{I_{c}}{I_{s}}$$(2)$${∆V}_{BE}=V_{BE2}-V_{BE1}=\frac{kT}{q}\ln8$$
where the symbols are defined as follows:

k represents the Boltzmann constant, with a value of k = 1.3806505 × 10^−23^ J/K.

q denotes the elementary charge of an electron, q = 1.6021892 × 10^−19^ C.

T is the absolute temperature in Kelvin, corresponding to a room temperature of 27 °C; thus, T = 300.15 K.

$I_{c}$ is the collector current of the transistor, and $I_{s}$ is its saturation current.

The forward voltages of the diodes in transistors $Q_{1}$ and $Q_{2}$ are denoted as $V_{BE1}$ and $V_{BE2}$, respectively. As the area of $Q_{1}$ is eight times that of $Q_{2}$, the voltage difference ${∆V}_{BE}$ is derived and presented in Equation (2).(3)$$I_{PTAT}=\frac{{∆V}_{BE}}{R_{0}}$$(4)$$I_{CTAT}=\frac{V_{BE2}}{R_{1}}$$(5)$$C*∆V=I*t=(I_{PTAT}+I_{CTAT})*\frac{N_{ref}}{CLK\text{\_}CS}$$(6)$$N_{ref}*(I_{PTAT}+I_{CTAT})=N_{out}*I_{PTAT}$$

The charging formula for capacitor $C_{1}$ is presented in Equation (5). Here, CLK_CS serves as a reference clock for counting operations, and $N_{ref}$ represents the reference count value. The charging and discharging behavior of capacitor $C_{1}$ is illustrated in Figure 1.



(7)
$$N_{out}=N_{ref}*\left(1+\frac{I_{CTAT}}{I_{PTAT}}\right)=N_{ref}*\left(1+\frac{R_{0}}{R_{1}\ln8}\ln\frac{I_{c}}{I_{s}}\right)$$



Equation (7) illustrates the effect of nonlinearity in the positive temperature coefficient. As temperature increases, $I_{CTAT}$ decreases while $I_{PTAT}$ increases, resulting in a decrease in $N_{out}$.(8)$$I_{c}=\frac{kT}{qR_{0}}\ln8=aT$$(9)$$I_{s}\propto \mu kTn_{i}^{2}={bT}^{\frac{5}{2}}e^{\frac{-Eg}{kT}}$$

Through the current mirror, the value of $I_{c}$ is designed to match the positive-temperature-coefficient current generated on resistor $R_{0}$, as shown in Equation (8). Additionally, the temperature-independent coefficient is simplified to a constant denoted by a. In Equation (9), the parameters are defined as follows:

μ represents the carrier mobility in the semiconductor, with $\mu \propto \mu_{0}T^{-\frac{3}{2}}$.

$n_{i}$ is the intrinsic carrier concentration of silicon, where $n_{i}^{2}\propto T^{3}e^{\frac{-Eg}{kT}}$.

b is the temperature-independent scaling factor after subtraction.

Eg is the energy of the bandgap of silicon, Eg = 1.12 eV.(10)$$\ln I_{s}=\ln b+\frac{5}{2}\ln T-\frac{Eg}{kT}$$(11)$$\ln\frac{I_{c}}{I_{s}}=\ln a+\ln T-\ln I_{s}=-\frac{3}{2}\ln T+\frac{Eg}{kT}+\ln\frac{a}{b}$$(12)$$N_{out}=N_{ref}*\left(1+\frac{R_{0}}{R_{1}\ln8}(-\frac{3}{2}\ln T+\frac{Eg}{kT}+\ln\frac{a}{b})\right)$$(13)$$N_{out}=-\frac{3R_{0}N_{ref}}{2R_{1}\ln8}\ln T+\frac{Eg}{kT}+N_{ref}+\frac{R_{0}N_{ref}}{R_{1}\ln8}\ln\frac{a}{b}$$(14)$$N_{\mathrm{out}}=\mathrm{Aln}T+\frac{B}{T}+C$$

Substituting Equation (11) into Equation (7) yields Equation (12); expanding Equation (12) results in Equation (13). Equation (14) is derived by simplifying the temperature-independent coefficients in Equation (13), where A, B, and C represent the temperature-independent coefficients following the simplification.

The charging current is decomposed into components with positive and negative temperature coefficients, where the temperature dependence cannot be fully canceled. Initially, the specific values of the positive and negative temperature coefficients can be ignored, as these vary across individual chips based on empirical measurements. These coefficients represent the calibration parameters unique to each chip. By this approach, we derive the temperature-to-digital conversion formula. It is observed that $\frac{B}{T}$ corresponds to one-point calibration, a commonly utilized method. Incorporating the coefficient C leads to two-point calibration, and the term AlnT introduces a nonlinear temperature dependence.

This calibration formula is applicable to temperature-dependent measurement curves characterized by a decreasing slope as temperature increases. Specifically, if a temperature sensor’s output response exhibits a decreasing slope with rising temperature, this calibration methodology can be employed to achieve accurate adjustment.

## 3. Circuit Principle

The proposed temperature sensor’s architecture is depicted in Figure 2. The overall circuit is divided into a startup circuit, positive-temperature current-generating circuit, negative-temperature current-generating circuit, current mirror circuit, and ADC circuit. In addition, there are control signal generation circuits and configuration circuitry for critical currents. The core element is the current-generation stage of the VPTAT, designed using a conventional bandgap reference circuit. A high-gain operational amplifier provides negative feedback, ensuring voltage equilibrium between VIP and VIN. The positive temperature coefficient ${∆V}_{BE}$ is generated across a zero-temperature-coefficient resistor $R_{0}$, producing a proportional positive-temperature-coefficient current. Simultaneously, the $V_{BE}$ voltage of transistor $Q_{2}$ induces a negative-temperature current across the zero-temperature resistor $R_{1}$ through buffering. Matching $R_{0}$ and $\;R_{1}$ minimizes resistance-induced temperature errors. All mirror nodes in the circuit are added to the filter capacitor and the initial pull-up or pull-down control, so the system has a stable initial state and working state.

Table 1 and Figure 3 provide a timing depiction of the voltages at critical nodes in the circuit, as well as a timing depiction of the critical registers inside the counter. The en_temp_delay signal is used to control the switch S_0_ to establish the initial state of the system. The signal S_ctrl is used to control the charging and discharging of the integrated capacitor by switches S_1_ and S_2_.

The following describes the selection of the operating voltage of the temperature sensor. In the preparation phase for the positive- and negative-temperature currents in the circuit core, switch S_0_ is closed, allowing the current to pre-charge the integrated capacitor. Once the reference voltage approaches 200 mV, the comparator reverses, transitioning the system out of the preparation phase and initiating the double integration logic. The reference voltage is determined based on the operating range of the NMOS discharge transistor MN_7_. The maximum voltage on the capacitor during the charging phase is determined by the operating range of the PMOS transistors MP_11_ and MP_13_. The drain–source voltage ($V_{DS}$) of both PMOS and NMOS transistors is set to approximately 200 mV. The voltage increase during the charging phase is 300 mV, with the voltage for both the charging and discharging sections theoretically reaching 700 mV, accounting for the deviations in the low-dropout regulator (LDO). The base-emitter voltage ($V_{BE}$) of the core is around 500 mV, and the $V_{DS}$ of the similarly designed PMOS transistors MP_1_ and MP_2_ is approximately 200 mV. A margin of about 100 mV is reserved, and the operating voltage of the temperature sensor is set to 0.8 V.

The following describes the working logic of the current-input dual-integral ADC. Initially, switch S_1_ is turned on, charging the capacitor within the specified time period T_1_. After 512 reference clock cycles are completed, switch S_1_ is turned off, ending the charging phase, and switch S_2_ is activated to initiate the discharge phase. When the voltage across the capacitor drops to the reference voltage, the comparator output transitions from 1 to 0. The digital logic then records the discharge time, denoted as T_2_, which corresponds to N reference clock cycles. This value of N is output by the digital logic to complete the temperature-to-digital conversion. To mitigate startup errors, the second count value N is used as the digital output for a single conversion, resulting in a conversion time of approximately 6 ms. The conversion accuracy of the ADC can be further enhanced by increasing the base count value and reference clock frequency. However, the design of the conversion time, reference clock frequency, and base count value of 512 is based on practical application considerations, specifically optimizing the energy efficiency and temperature accuracy during communication with the reader, while also accounting for the return time and power consumption of the write instruction as specified in the Gen2 protocol.

The temperature sensor consumes 1 uA at a room temperature of 27 °C and supplies 0.8 V. To achieve robust functionality under PVT (Process–Voltage–Temperature) variations, the current mirror is designed to operate at a minimum mirrored current of 5 nA. The charging current is set at 10 nA and the discharge current at 5 nA. A high-gain folded cascode operational amplifier with self-generated internal biasing ensures circuit stability and provides bias voltages to other operational amplifiers, maintaining a total power consumption of 100 nA. Given the circuit’s ultra-low power consumption, high-temperature device leakage effects must be accounted for, necessitating temperature sensor calibration. The circuit layout is shown in Figure 4, with an overall area of 214 μm × 258 μm.

The overall architecture of the UHF (ultra-high-frequency) RFID chip is depicted in Figure 5. Within the RF section, the rectification module converts the ultra-high-frequency signal into the chip’s power supply, VDD. Upon receiving VDD, the analog section activates the BIAS module to generate the reference current required for the chip’s operation. When VDD reaches the power-on threshold defined by the Power-On Reset (POR) module, it outputs a high level, enabling the low-dropout regulator (LDO) and Clock Generation (CLK) modules to initiate normal operation. This process generates the digital voltage and reference clock necessary for the other chip modules. In this stage, the POR signal, clock signal, and digital power supply needed for the operation of the RFID Baseband (RFID_BB) module are fully prepared, allowing the RFID_BB to begin its initialization. During initialization, the RFID_BB retrieves system configuration parameters from the Electrically Erasable Programmable Read-Only Memory (EEPROM) to configure the RFID system. Once the initialization of RFID_BB is complete, the RFID system enters standby mode, awaiting commands from the reader. At this point, a write command is sent through the reader to activate the temperature sensor module, which measures the ambient temperature and returns the quantized temperature value to the reader. Since the RFID chip operates on radio-frequency energy, reducing overall power consumption is crucial for enhancing the chip’s sensitivity. Therefore, the described startup scheme has been adopted to minimize power usage. Additionally, once the chip enters standby mode, the energy consumed during command reception and data transmission is reduced to half of the original value. Considering the temperature sensor’s measurement time and the command time specified by the Gen2 protocol, the write command is employed to trigger the temperature sensor for temperature measurement, ensuring that the power consumption of the temperature sensor during operation remains lower than the total power consumption of the chip during command reception and data return.

The use of the RFID temperature measurement function is depicted in Figure 6. When the reader identifies a tag and sets the tag in open or secured state, data 0000 h is written to the address 81 h to open the sensor, and then the read command is sent to obtain the data of the temperature sensor. Table 2, Table 3, Table 4 and Table 5 provide the description of the write and read command. In this work, several key abbreviations utilized within the command are presented below:

QueryRep: Query repeat;

ACK: Acknowledge character;

Req_RN: Request random or pseudo-random number;

MemBank: Memory bank;

WordPtr: Word pointer;

RN: Random or pseudo-random number;

CRC-16: 16-bit Cyclic redundancy check.

## 4. Results

The chip was tested using a QR4453 reader controlled by a PC program, with the chip placed in a constant-temperature thermostat. As shown in Figure 7 and Figure 8, the system architecture and test environment for the chip evaluation are illustrated. The chip was selected and positioned in the constant-temperature thermostat, with the temperature measurement function activated via the program-controlled reader on the PC. The temperature readings from the chip were recorded at various set temperatures. During testing, the chips were placed in a custom-designed closed fixture, with each fixture capable of holding up to 96 chips simultaneously.

Figure 9 illustrates the QR4453 reader, a high-performance UHF fixed reader based on the Impinj R2000 module, which supports the ISO 18000-6C air interface protocol. Table 6 provides the product features of the QR4453 reader.

Five chips exhibiting a temperature error exceeding ±1 °C after one-point calibration were selected for further analysis. Table 7 presents the temperature errors of these five chips following one-point, two-point, and three-point calibration. As the number of calibration points increases, the linearity of the calibration improves. The temperature curve is effectively divided into multiple linear segments, reducing the overall temperature error with additional calibration points. However, implementing this approach in large-scale production presents significant challenges. Since a single wafer contains tens of thousands of chips, with each requiring testing at multiple temperature points, repeated measurements increase wafer stress and potential damage. Additionally, the cost of testing escalates with the number of calibration iterations, making extensive calibration impractical for mass production.

Figure 13 illustrates the slope of the five tested chips as a function of temperature, which aligns with the observed trends summarized in this study. The charging current exhibits a nonlinear positive temperature coefficient rather than a zero-temperature coefficient. Consequently, as the temperature increases, the slope decreases.

To further evaluate the effectiveness of the nonlinear calibration, the estimated temperature error was calculated using the nonlinear calibration formula, with the results presented in Figure 14.

Tagformance was used to evaluate the sensitivity of the RFID tag with the temperature measurement function enabled. Figure 15 shows that the average sensitivity was approximately −15 dBm.

Figure 16 illustrates the RFID tag using a QR4453 reader controlled by a PC program and illustrates the shape of the signal that controls the RFID modulator circuit during the temperature measurement. First, the tag inventory should be selected through the PC program, the target tag should be selected, Get-Tag-Temperature should be clicked, and then the obtained temperature value (0392) is displayed in the program interface. RFID commands and reply signals are captured by using an RF-coupled test board which connects to an oscilloscope.

Compared with the calibration scheme of high- and low-power temperature sensors proposed in this paper, in order to achieve the requirement of temperature errors within ±1 °C of −30 °C~90 °C, three or more traditional linear calibration schemes are required, and the calibration error is strongly correlated with the change in slope of the temperature measurement curve. The three-point nonlinear calibration scheme explains the cause of the high-temperature nonlinearity from the circuit principle, and verifies the accuracy of the scheme through the measurement results, making it very suitable for waveforms where the slope decreases with the increase in temperature. Table 8 presents a comparison of the power consumption and temperature error of the proposed temperature sensor with other existing temperature sensors [6,7,8,9,10,11].

The temperature calibration method proposed in this paper, implemented at the reader end, effectively addresses the challenges of power consumption and area constraints associated with on-chip internal calibration while enhancing calibration accuracy in practical applications. Since the operational energy of the chip is derived from the RF field strength generated by the reader, embedding nonlinear calibration within the chip would lead to excessive power and area consumption, thereby reducing the temperature monitoring range and system reliability. This outcome would contradict the fundamental objective of achieving high-efficiency temperature measurement, making it unsuitable for practical use.

To reduce these limitations, a computational algorithm is integrated into the RFID mobile terminal (UHF handheld), employing a dichotomous approach to determine the temperature value. This value is subsequently compared with the sensor reading, enabling the control of comparison accuracy to ensure that the temperature error falls within the acceptable range for practical applications.

Figure 17 illustrates the specific implementation flow. The RFID mobile terminal sends read commands to obtain sensor data M and three calibration data points, e.g., N0(@−30 °C), N1(@27 °C), and N2(@90 °C). Then, coefficients A, B, and C are calculated from these calibration data. By comparing M with N0, N1, and N2 to determine the temperature range of M, the initial values Ta and Tb of the dichotomy are obtained. The temperature is obtained by successive approximation.

## 5. Conclusions

The ultra-low-power, high-precision temperature sensor proposed in this paper meets the temperature measurement requirements of passive IoT systems, achieving a power consumption of 0.8 µW and an inaccuracy of +0.6/−1 °C within the temperature range of −30 °C to 90 °C. The sensitivity of the UHF RFID tag was evaluated using Tagformance, and the average sensitivity of the UHF RFID tag with the temperature measurement function enabled was measured at −15 dBm.

This paper analyzes the causes of the nonlinear error in the temperature sensor and verifies the source of temperature deviations through experimental data. The charging current exhibits a nonlinear positive temperature coefficient rather than a zero-temperature coefficient. Based on this characteristic, a mathematical equation describing the nonlinear curve is derived, leading to the development of a three-point nonlinear calibration scheme. This calibration formula is suitable for temperature measurement curves where the slope slows down as the temperature increases.

To address calibration challenges, the proposed method integrates the calibration algorithm into the reader, effectively correcting the temperature sensor’s nonlinear error. This approach reduces the temperature error to ±1 °C while maintaining a reasonable calibration cost. Due to the extensive chip testing required, measurements above 125 °C were not conducted; however, data obtained within the tested range of −30 °C to 90 °C demonstrate the effectiveness of the proposed method.

Future work will focus on integrating a compensation circuit within the chip to correct the nonlinearity in the high-temperature range, thereby improving measurement linearity. This enhancement is expected to achieve an inaccuracy of +1/−1 °C across a broader temperature range of −40 °C to 125 °C with a one-point calibration approach. In practical applications, once the temperature measurement function is enabled, the calibrated temperature value can be directly transmitted to the reader, enhancing the generalizability of RFID-based temperature sensing.

## Figures and Tables

**Figure 1 sensors-25-02911-f001:**
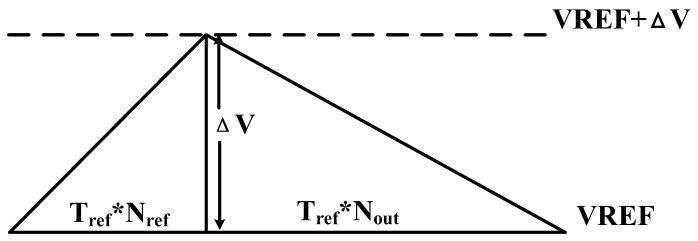
Voltage changes on the capacitor $C_{1}$ ($T_{ref}=\frac{1}{CLK\text{\_}CS}$).

**Figure 2 sensors-25-02911-f002:**
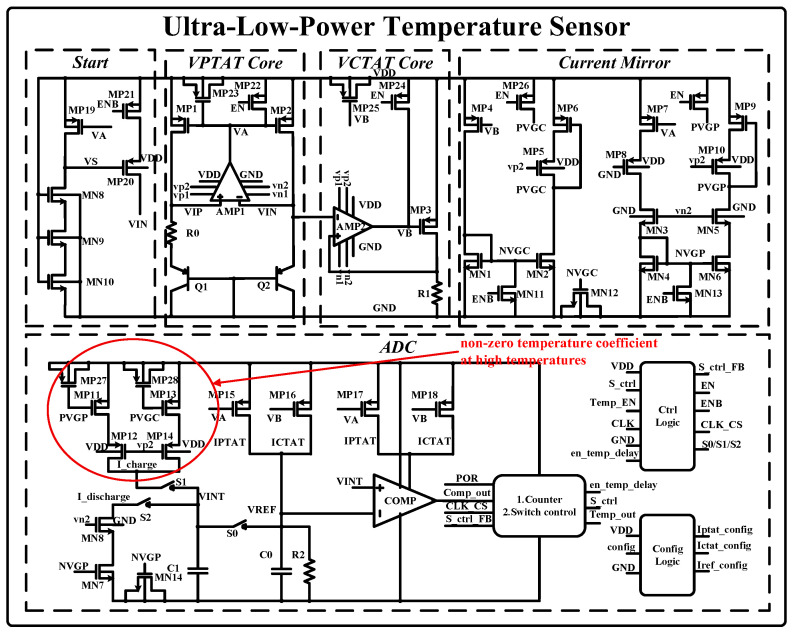
Schematic diagram of temperature sensor.

**Figure 3 sensors-25-02911-f003:**
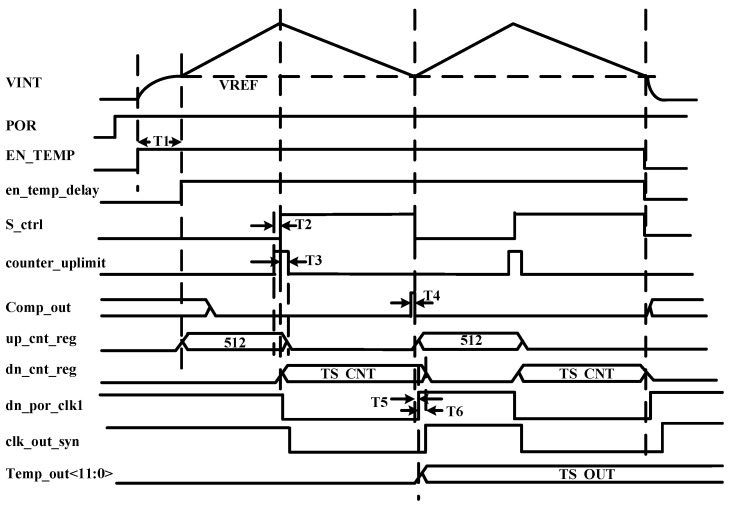
The timing of important signals.

**Figure 4 sensors-25-02911-f004:**
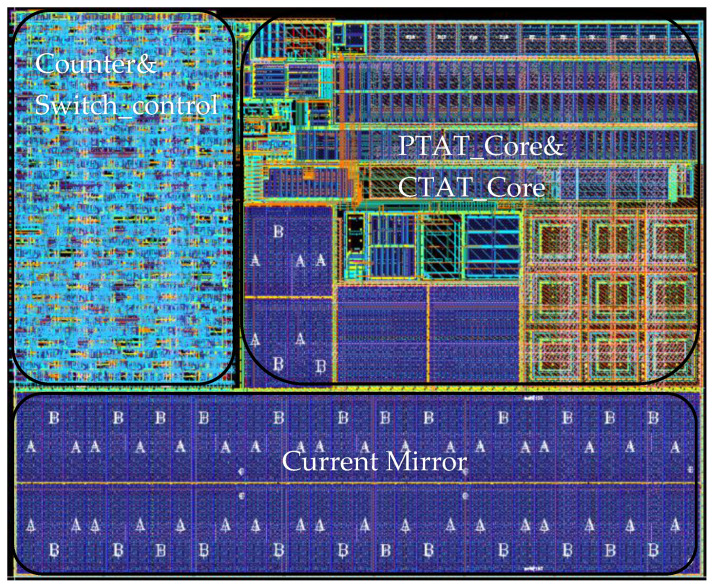
The layout of the temperature sensor.

**Figure 5 sensors-25-02911-f005:**
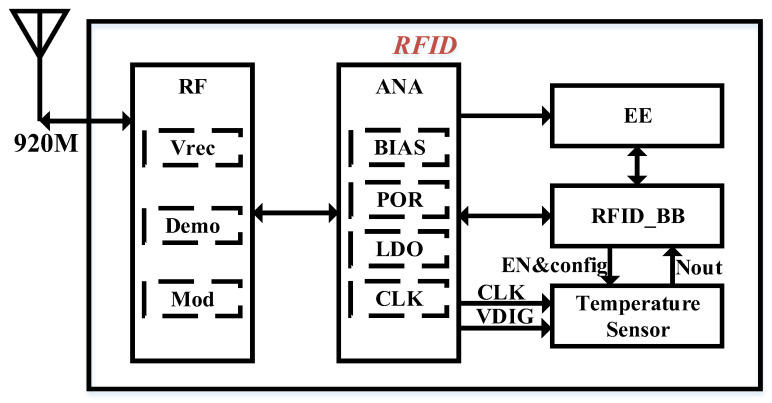
The structure of the UHF RFID.

**Figure 6 sensors-25-02911-f006:**
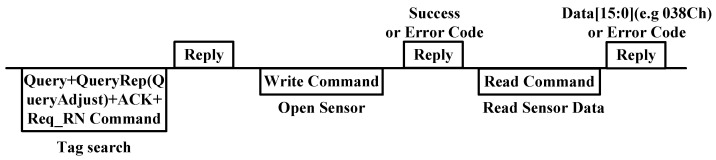
Temperature sensor control flow.

**Figure 7 sensors-25-02911-f007:**
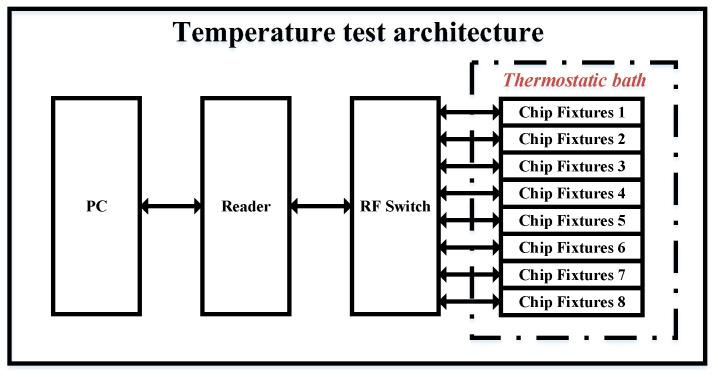
Temperature test architecture.

**Figure 8 sensors-25-02911-f008:**
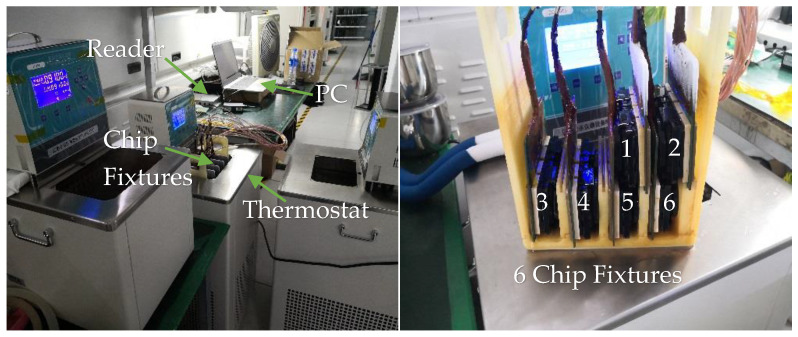
Temperature test environment.

**Figure 9 sensors-25-02911-f009:**
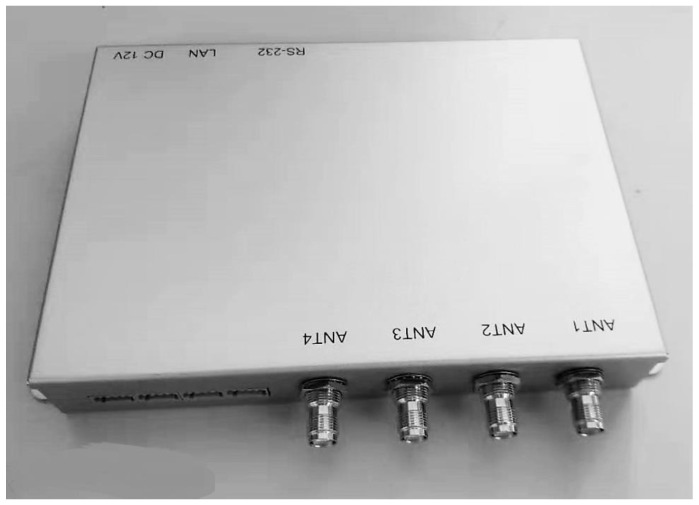
The QR4453 reader.

**Figure 10 sensors-25-02911-f010:**
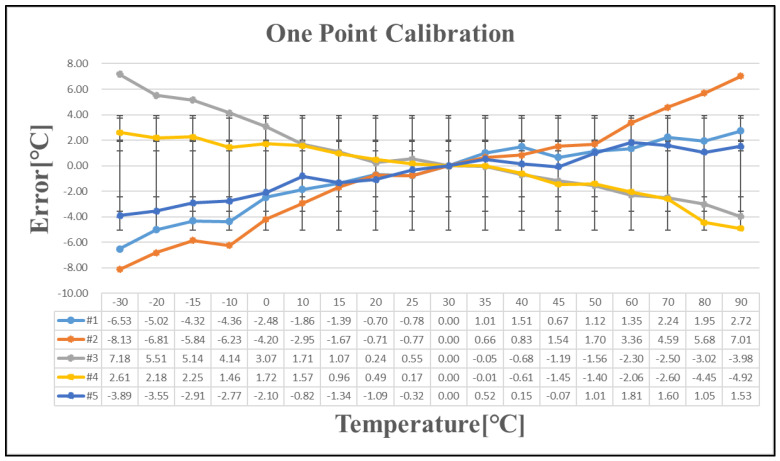
Temperature error after one-point calibration.

**Figure 11 sensors-25-02911-f011:**
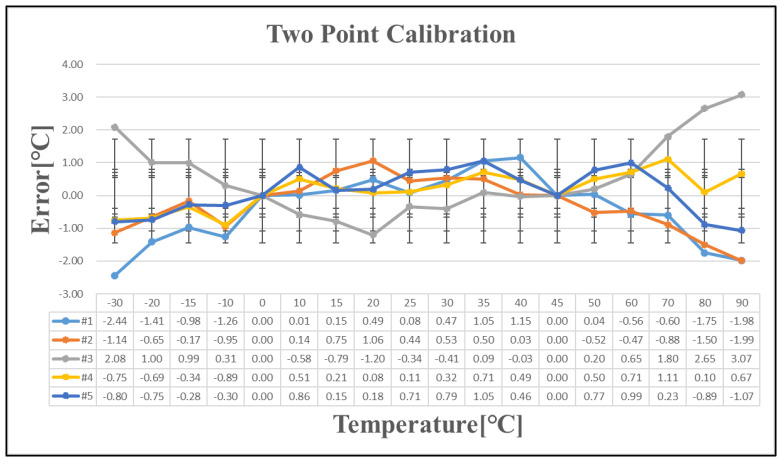
Temperature error after two-point calibration.

**Figure 12 sensors-25-02911-f012:**
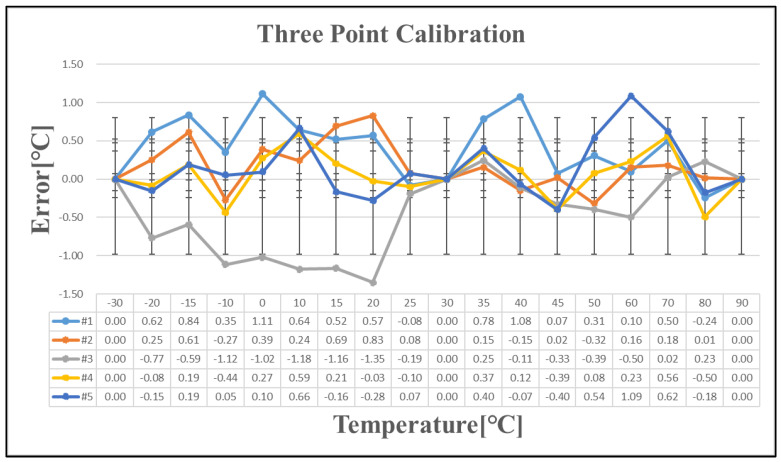
Temperature error after three-point calibration.

**Figure 13 sensors-25-02911-f013:**
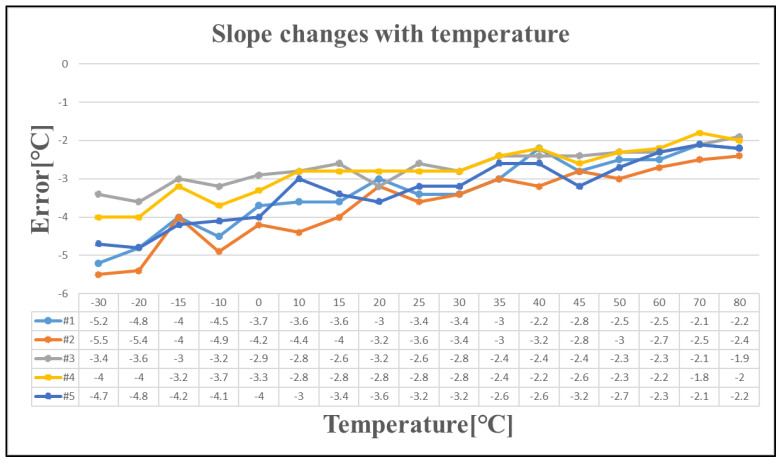
The slope changes with temperature.

**Figure 14 sensors-25-02911-f014:**
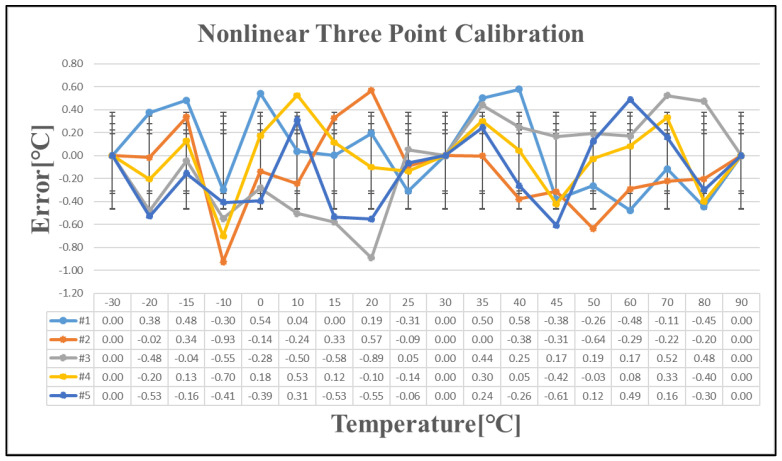
Temperature error after nonlinear three-point calibration.

**Figure 15 sensors-25-02911-f015:**
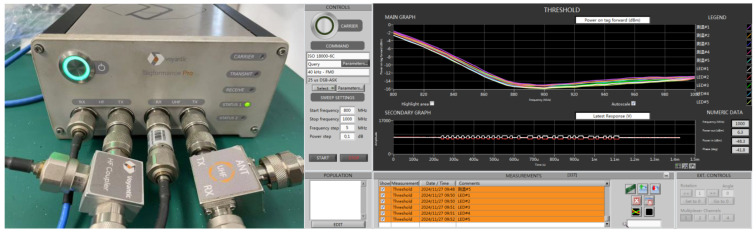
RFID tag sensitivity test with temperature sensor turned on.

**Figure 16 sensors-25-02911-f016:**
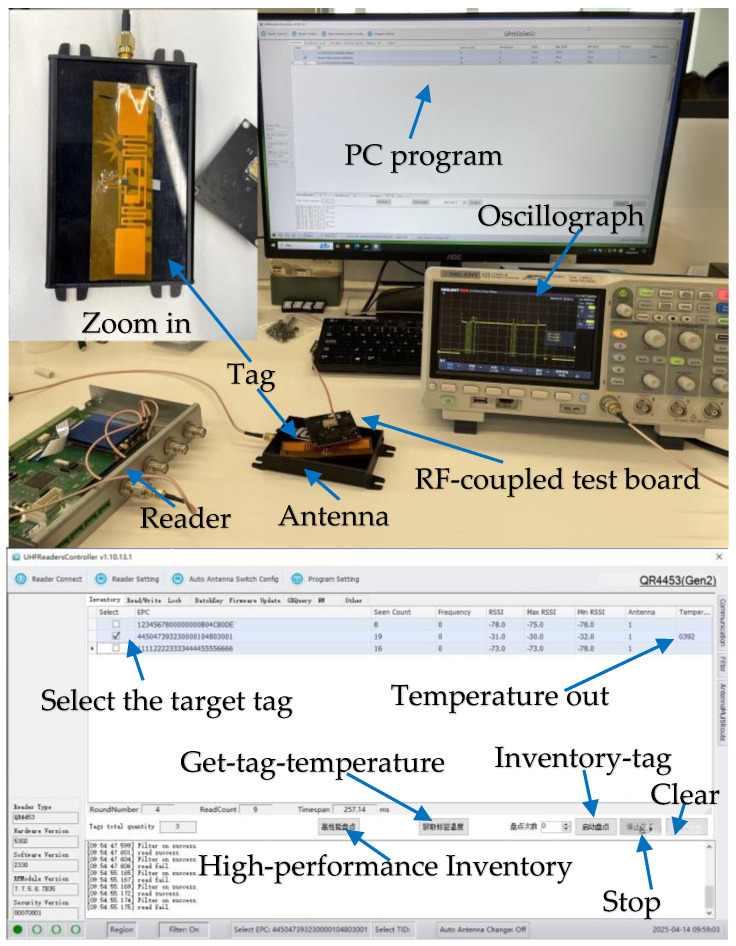
RFID tag temperature measurement command process.

**Figure 17 sensors-25-02911-f017:**
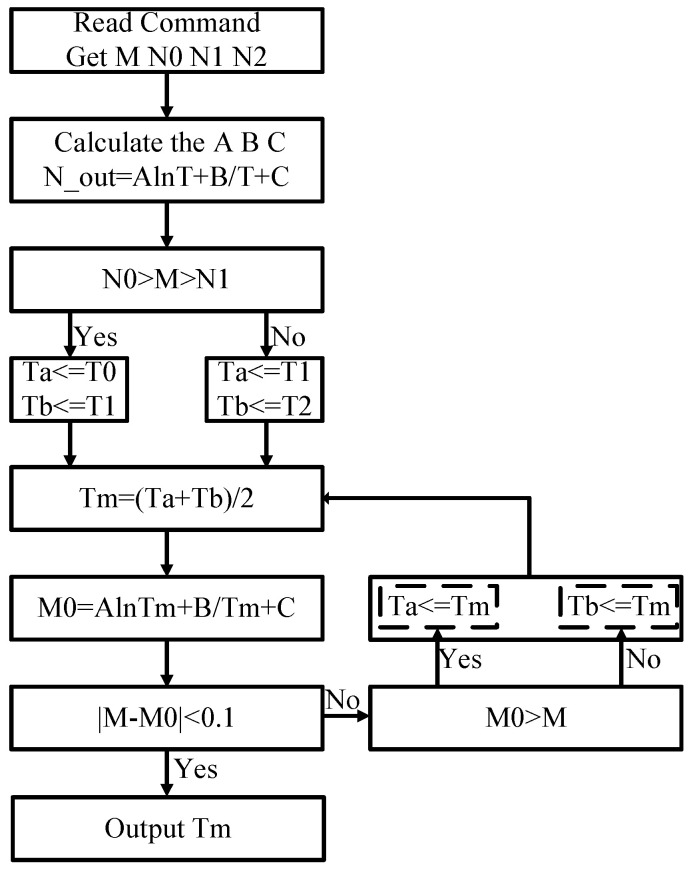
The flow of the reader to output temperature.

**Table 1 sensors-25-02911-t001:** The description of important signal timings.

Symbol	Description	Min	Typ	Max
T1	For the enable signal EN_TEMP, delay T_1_ time adopts the enable time of the internal digital circuit, which is also the preparation time of the analog circuit		128/CLK [ns]	
T2	The time when up_cnt_reg reaches 512 counts so that counter_uplimt is high and S_ctrl is high		2 [ns]	
T3	The time when S_ctrl is high to reset Up_cnt_reg to low counter_uplimt			1 [ns]
T4	The time when the comparator output is high to set S_ctrl is low		2 [ns]	
T5	The time from the S_ctrl falling edge to the dn_por_clk1 rising edge	2 [ns]		1/CLK [ns]
T6	The time from the dn_por_clk1 rising edge to reset dn_cnt_reg		1/CLK [ns]	

**Table 2 sensors-25-02911-t002:** Write command.

	Command	MemBank	WordPtr	Data	RN	CRC-16
bits	8	2	EBV	16	16	16
Description	11000011	11	81 h	0000 h	Handle	-

**Table 3 sensors-25-02911-t003:** Write successful return command.

	Header	RN	CRC-16
bits	1	16	16
Description	0	Handle	-

**Table 4 sensors-25-02911-t004:** Read command.

	Command	MemBank	WordPtr	WordCount	RN	CRC-16
bits	8	2	EBV	8	16	16
Description	11000011	11	81 h	1	Handle	-

**Table 5 sensors-25-02911-t005:** Read successful return command.

	Header	Memory Words	RN	CRC-16
bits	1	Variable	16	16
Description	0	Data	Handle	-

**Table 6 sensors-25-02911-t006:** QR4453 reader product features.

Air Interface Protocol Supports	ISO18000-6C (EPC C1G2) Protocol	
Electrical performance parameters	Transmit power	0–33 [dBm], step 1 [dB]
	Operating frequency	902~928 [MHz]
	Read sensitivity	≤−70 [dBm]@30 [dBm]
	Reading distance	>10 [m] (8 [dBi] antenna)
	Label processing speed	>200 [pcs/sec]
	Communication interface	RS232, RJ45
	Data interface	SDIO/GPIO
	Power supply	12 [V]/3 [A]
	System power consumption	1.3 [A] × 5 [V] peak@30 [dBm]
Environmental indicators	Operating temperature	−20 [°C]~55 [°C]
	Storage temperature	−40 [°C]~80 [°C]
Mechanical Characteristics	Dimensions	203 [mm] × 150 [mm] × 48 [mm]
	IP protection	IP40

**Table 7 sensors-25-02911-t007:** Temperature error after linear calibration.

Sensing Error	Linear Calibration	Figure
+7/−8 [°C]	One-point	Figure 10
+3/−2.5 [°C]	Two-point	Figure 11
+1.2/−1.4 [°C]	Three-point	Figure 12

**Table 8 sensors-25-02911-t008:** Comparison of performance on several temperature sensors.

Contrast	Process	Power	Temperature Sensing Range	Sensing Error	Calibration	Sensitivity
Reference [6]	0.18 um CMOS	13.2 [uW]	−20 [°C] to 50 [°C]	+0.8/−1 [°C]	One-point	Not applicable
Reference [7]	0.18 um CMOS	2.4 [uW]	−20 [°C] to 30 [°C]	+0.8/−0.8 [°C]	One-point	−6 [dBm]
Reference [8]	Not applicable	9.6 [uW]	0 [°C] to 100 [°C]	+2.5/−2 [°C]	One-point	Not applicable
Reference [9]	0.18 um CMOS	2.0 [uW]	0 [°C] to 100 [°C]	+1/−1.8 [°C]	Not applicable	−11 [dBm]
Reference [10]	0.18 um CMOS	0.35 [uW]	−30 [°C] to 60 [°C]	+1.5/−1.5 [°C]	One-point	Not applicable
Reference [11]	0.18 um CMOS	0.12 [uW]	−10 [°C] to 30 [°C]	+1/−0.8 [°C]	Two-point	0 [dBm]
This work	0.18 um CMOS	0.8 [uW]	−30 [°C] to 90 [°C]	+0.6/−1 [°C]	Three-point	−15 [dBm]

## Data Availability

No new data were created.

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
