# Peer review of "An Ultra-Low-Power High-Precision Temperature Sensor Using Nonlinear Calibration with an Inaccuracy of +0.6/−1 °C from −30 °C to 90 °C for RFID Applications"

_sensors, 2025, doi:10.3390/s25092911_

Round 1
Reviewer 1 Report
Comments and Suggestions for Authors
This study presents a tri-stage nonlinear compensation method for ultra-low-energy, high-accuracy temperature sensing systems, tackling the ±1℃ threshold exceedance observed in 0.8μW RFID-based thermometric devices. The developed strategy enhances conventional linear compensation frameworks by incorporating a temperature-sensitive nonlinear correction factor, thereby addressing inherent sensor output nonlinearities. To optimize implementation efficiency, a reader-integrated compensation architecture is designed alongside a bisection method-driven calibration protocol for rapid thermal parameter tuning. Validation tests demonstrate error suppression from initial ±8℃ variations to a tightly controlled -1℃~+0.6℃ range across five sensor prototypes post-calibration.
Some concerns:
1. This methodology is only applicable to the sensor model in the study. For other nonlinear temperature sensors, three reference points may not suffice to determine temperatures across all ranges, limiting the paper's contribution as a temperature calibration method.
2. While RFID temperature sensors hold practical significance, numerous relevant studies have emerged post-2020. What is the technical novelty of this work compared to recent advances ?
3. The claim about retrieving measured temperatures via the write command may be incorrect. Per the C1G2 protocol, this command does not trigger tag data replies. Clarify the implementation principles in alignment with the protocol.
4. The method still requires per-tag registration – propose potential optimizations to streamline this process.
5. The manuscript requires improved clarity and linguistic fluency.
Comments on the Quality of English LanguageThe writing needs to be improved.
Author Response
comments1:This methodology is only applicable to the sensor model in the study. For other nonlinear temperature sensors, three reference points may not suffice to determine temperatures across all ranges, limiting the paper's contribution as a temperature calibration method.
Reply: This calibration formula is generally applicable to the curve of slope slowing, according to the curve presented by the actual test value of other temperature sensors, if there is a situation that the slope slows down with the increase of temperature, it can be calibrated with this calibration scheme.
While RFID temperature sensors hold practical significance, numerous relevant studies have emerged post-2020. What is the technical novelty of this work compared to recent advances ?
Reply: Compared with other RFID temperature sensors, the circuit in this study consumes less power, has a smaller area, and is more energy efficient. The nonlinear three-point calibration fusion scheme proposed in this paper can achieve a wider temperature measurement range and smaller temperature error
The claim about retrieving measured temperatures via the write command may be incorrect. Per the C1G2 protocol, this command does not trigger tag data replies. Clarify the implementation principles in alignment with the protocol.
Reply: Include a specific description of the instructions and photos of the communication process
Reviewer 2 Report
Comments and Suggestions for Authors
This paper presents a method for reducing the errors of CMOS temperature sensors attached to RFID circuits by using a correction algorithm in the signal processing circuitry that is part of the RFID reader structure. From the graphical material presented in the paper it appears that the integrated circuit that allows the temperature-to-digital signal conversion is of high complexity, but the technical content does not bring enough novelty compared to the previous version presented at the ASICON 2024 Conference, but only extends, to some extent, this version. Although a new set of photographs is presented in addition to the ASICON 2024 version, they do not provide technically or scientifically relevant information.
The Introduction chapter is extremely short and does not present the usual comparison between the state-of-art and the solution described in the paper submitted for review, but only comparisons between several CMOS sensor designs (references [4] - [9], excepting [6]). Thus, the wiring diagrams, IC layout, formulas connecting different electrical quantities in the integrated circuit, plot of the measurement errors after calibrating the RFID reader at a single point were already used in the 2024 version of the paper.
The shortcomings listed above make it impossible to evaluate the paper according to the criteria set by the Sensors journal for the acceptance of a paper in terms of (i) appropriateness of the way the authors have designed the research model; (ii) adequate description of the methods; (iii) clarity of the results presented; (iv) conclusions supported by the results.
If the Editor decides to accept the work with major revisions, the authors should consider the following:
- Revising the Introduction chapter by adding comments on the following topics: (i) currently used solutions for integrating sensors, temperature or other models, into the RFID structure, including advantages and disadvantages; (ii) comparisons of the range of measured temperatures, complexity of realization technologies, dimensions; (iii) external factors affecting the accuracy of the sensors depending on the construction solution, (iv) anticipated application limits of the proposed solution. These descriptions and/or comparisons are a must for the readers.
- Since the Bibliography chapter currently has only nine bibliographical references, of which [3] is the very version of the paper presented by the same authors at the ASICON 2024 Conference, titled "An ultra-low-power temperature sensor with an inaccuracy of +0.6/-1℃ from -30℃ to 90℃", more recent references need to be added for a full and fair comparison of the advantages and disadvantages of the sensor model proposed by the authors compared to temperature sensors whose operation is based on other principles. Excepting [3], the other references are quite old: seven were published between 2006 and 2016 and only one in 2018.
- Photograph of the RFID incorporating the sensor realized by the authors and the shape of the signal that controls the RFID modulator circuit during a complete cycle will be added.
- A few comments are needed on the components and/or devices in the RFID architecture that limit the maximum operating temperature of the system.
Further comments:
- It is necessary to associate an acronym with a particular group of words from the first appearance of that group in the text. In line 41, the acronym ADC is used for the first time; it is recommended that its meaning "analogue-to-digital converter" be included in the text, even if it is well known. In line 113, the acronym PVT appears, but it is not accompanied by an extension of the terms to which it refers.
- Row 83: replace "integral capacitor" with "integrated capacitor".
- Row 136: replace "transitions into standby mode" with "enters standby mode".
Author Response
comments1: (i) currently used solutions for integrating sensors, temperature or other models, into the RFID structure, including advantages and disadvantages; (ii) comparisons of the range of measured temperatures, complexity of realization technologies, dimensions; (iii) external factors affecting the accuracy of the sensors depending on the construction solution, (iv) anticipated application limits of the proposed solution. These descriptions and/or comparisons are a must for the readers.
Since the Bibliography chapter currently has only nine bibliographical references, of which [3] is the very version of the paper presented by the same authors at the ASICON 2024 Conference, titled "An ultra-low-power temperature sensor with an inaccuracy of +0.6/-1℃ from -30℃ to 90℃", more recent references need to be added for a full and fair comparison of the advantages and disadvantages of the sensor model proposed by the authors compared to temperature sensors whose operation is based on other principles. Excepting [3], the other references are quite old: seven were published between 2006 and 2016 and only one in 2018.
Reply: Your suggestion is very good, and the introduction has been revised in the light of your proposed amendments
Photograph of the RFID incorporating the sensor realized by the authors and the shape of the signal that controls the RFID modulator circuit during a complete cycle will be added.
Reply:A specific RFID communication process diagram has been added to the paper
A few comments are needed on the components and/or devices in the RFID architecture that limit the maximum operating temperature of the system.
Further comments:
It is necessary to associate an acronym with a particular group of words from the first appearance of that group in the text. In line 41, the acronym ADC is used for the first time; it is recommended that its meaning "analogue-to-digital converter" be included in the text, even if it is well known. In line 113, the acronym PVT appears, but it is not accompanied by an extension of the terms to which it refers.
Row 83: replace "integral capacitor" with "integrated capacitor".
Row 136: replace "transitions into standby mode" with "enters standby mode".
Reply: Thanks for pointing out my bug, it has been modified
Reviewer 3 Report
Comments and Suggestions for Authors
See attached file

Author Response
comments1:The hardware description of the chip is detailed and technically sound. However, the level of detail devoted to the algorithm does not match that of the hardware section. Specifically, the equations presented on page 2 should be discussed more thoroughly to clarify how the nonlinear term is introduced and how it is computed by the algorithm. Additional commentary on the formulas and clarification of acronyms would further enhance readability and understanding.
Reply:Your suggestion is good, it is true that the description of the hardware part and the calculation formula was not enough before, it has been modified
The authors state (line 215) that the proposed approach requires fewer calibration points. However, the number of points used is still three, which is not necessarily fewer than in alternative solutions. In fact, as shown in Table 3, other chips in the comparison use an even smaller number of calibration points.
Reply:The meaning expressed earlier is ambiguous, and this part of the description has been modified
One of the stated advantages of the proposed method is the integration of the calibration procedure into the reader. However, the implementation details of this integration are not clearly presented. Including flowcharts or block diagrams would help illustrate the traditional calibration process versus the one proposed in this work. Additionally, the steps described between lines 228 and 231 should be expanded upon, as they appear to represent the core innovation of the paper.
Reply:A specific description of the 2-point calibration process has been added
In the final section, particularly between lines 220 and 246, the same concept is reiterated multiple times. This section would benefit from a more concise and focused presentation.
Reply:Thanks for pointing out the problem, revised
Units of measurement should be included in all figures, enclosed in square brackets, to improve clarity and adherence to standard scientific conventions.
Reply:Thanks for pointing out the problem, revised
Round 2
Reviewer 1 Report
Comments and Suggestions for Authors
The authors have addressed all of my concerns. The manuscript is now in good shape.
Author Response
Thank you for your valuable comments and suggestions in your previous review.
Reviewer 2 Report
Comments and Suggestions for Authors
- The Introduction chapter has been expanded, but without responding to the request for a comparison with other different temperature sensing elements and devices, such as thermocouples, resistors, thermistors, surface acoustic wave (SAW) devices, various reflective structures (backscattering, passive). Their presence in the paper is necessary in order to justify the technical solution that has been developed.
- Also, the Introduction is poorly written, with sections of text added by the authors without any regard for their logical order. For example, lines 59–62 should be deleted, as they repeat the final paragraph (lines 66–69). The final paragraphs also need to be reworded, as the current arrangement of ideas is unclear.
- The new section in Chapter 2, which introduces a set of mathematical relations, is carelessly worded: some of the symbols associated with terms and factors are missing subscripts and superscripts.
- A more detailed description of the signal timing constraints for the A/D converter operation listed in Table 1 should be added to lines 147–155 of Chapter 3.
- Definitions should be added to the text in lines 225-234 for the newly interoded acronyms, e.g. Acknowledgement for ACK, but also for Query, Query Rep, Req_RN, etc.
It is necessary to carefully revise the text from a grammatical point of view.
Author Response
Comment 1:The Introduction chapter has been expanded, but without responding to the request for a comparison with other different temperature sensing elements and devices, such as thermocouples, resistors, thermistors, surface acoustic wave (SAW) devices, various reflective structures (backscattering, passive). Their presence in the paper is necessary in order to justify the technical solution that has been developed.
Reply:We appreciate your suggestions and acknowledge that our previous analysis did not include commonly used off-chip temperature measurement methods such as thermocouples, thermistors, and surface acoustic wave (SAW) devices. This revision incorporates these methods, enhancing the comprehensiveness of our paper's introduction.
Comment 2:Also, the Introduction is poorly written, with sections of text added by the authors without any regard for their logical order. For example, lines 59–62 should be deleted, as they repeat the final paragraph (lines 66–69). The final paragraphs also need to be reworded, as the current arrangement of ideas is unclear.
Reply:Thank you for your suggestions. We acknowledge that the introduction was poorly written previously and have revised its structure to improve its flow and clarity.
Comment 3:The new section in Chapter 2, which introduces a set of mathematical relations, is carelessly worded: some of the symbols associated with terms and factors are missing subscripts and superscripts.
Reply:We appreciate your suggestions and have carefully reviewed and optimized this description to better convey the intended meaning.
Comment 4:A more detailed description of the signal timing constraints for the A/D converter operation listed in Table 1 should be added to lines 147–155 of Chapter 3.
Reply:Thank you for pointing this out. It was indeed an oversight to place the sequential description after the figure. We have modified it according to your suggestion to ensure better readability and logical coherence.
Comment 5:Definitions should be added to the text in lines 225-234 for the newly interoded acronyms, e.g. Acknowledgement for ACK, but also for Query, Query Rep, Req_RN, etc.
Reply:Thank you for your suggestions. We have added definitions for these abbreviations to improve the clarity and accessibility of the text.